



# Unprecedented Summer Hypoxia in Southern Cape Cod Bay: An Ecological Response to Regional Climate Change?

Malcolm E. Scully[1], W. Rockwell Geyer[1], David Borkman[2], Tracy L. Pugh[3], Amy Costa[4], and Owen C. Nichols[4]

[1]Applied Ocean Physics and Engineering, Woods Hole Oceanographic Institution, Woods Hole, MA, 02543, USA
[2]Rhode Island Department of Environmental Management, Providence, RI, 02908, USA
[3]Massachusetts Division of Marine Fisheries, New Bedford, MA, 02744, USA
[4]Center for Coastal Studies, Provincetown, MA, 02657, USA

*Correspondence to*: Malcolm E. Scully (mscully@whoi.edu)

**Abstract.** In late summer 2019 and 2020 bottom waters in southern Cape Cod Bay (CCB) became depleted in dissolved oxygen (DO), with documented benthic mortality in both years. Hypoxic conditions formed in relatively shallow water where the strong seasonal thermocline intersected the sea floor, both limiting vertical mixing and concentrating biological oxygen demand (BOD) over a very thin bottom boundary layer. In both 2019 and 2020, anomalously high sub-surface phytoplankton blooms were observed, and the biomass from these blooms provided the fuel to deplete sub-pycnocline waters
of DO. The increased chlorophyll fluorescence was accompanied by a corresponding decrease in sub-pycnocline nutrients, suggesting that prior to 2019 physical conditions were unfavorable for the utilization of these deep nutrients by the late summer phytoplankton community. It is hypothesized that significant alteration of physical conditions in CCB during late summer, which is the result of regional climate change, has favored the recent increase in sub-surface phytoplankton production. These changes include rapidly warming waters and significant shifts in summer wind direction, both of which
impact the intensity and vertical distribution of thermal stratification and vertical mixing within the water column. These changes in water column structure are not only more susceptible to hypoxia, but also have significant implications for phytoplankton dynamics, potentially allowing for intense late summer blooms of *Karenia mikimotoi*, a species new to the area. *K. mikimotoi* had not been detected in CCB or adjacent waters prior to 2017, however increasing cell densities have been reported in subsequent years, consistent with a rapidly changing ecosystem.

## 1 Introduction

The occurrence of coastal hypoxia is increasing worldwide, and this rapid expansion is often linked to anthropogenic nutrient inputs (Diaz, 2001). Hypoxia caused by eutrophication is generally found in coastal regions that receive large freshwater inputs and the associated terrestrial nutrients (Fennel and Testa, 2019). Low dissolved oxygen (DO) also occurs in coastal waters not directly impacted by large freshwater inputs. For example, hypoxia on continental shelves
can result from the upwelling of deep, nutrient-rich waters that enhance surface productivity and deliver oxygen-poor bottom



waters to the shelf (Grantham et al. 2004). Long-term changes in hypoxia in these environments may be associated with long-term changes in the wind forcing, and the associated changes in upwelling dynamics (Garcia-Reyes et al. 2015). Even in regions with high anthropogenic nutrient loading, variations in hypoxia are strongly influenced by variations in physical forcing, so that both large-scale and regional climate variability can have important impacts on the occurrence of hypoxia in

coastal waters (e.g., Scully, 2010). Processes such as wind forcing, water temperature and vertical density stratification all play important roles in modulating bottom DO in a wide range of coastal environments (Wilson et al. 2008; Forrest et al. 2011; Yu et al. 2015; Scully, 2016). These physical processes control DO saturation (temperature) and horizontal advection and vertical mixing, which have direct impacts on the solubility and physical transport of DO, and hence hypoxia.

In addition to these direct impacts on physical transport, changes to the physical environment can significantly alter

phytoplankton dynamics, which can also lead to hypoxia. Phytoplankton are the base of the marine food web, providing the organic carbon that ultimately drives hypoxia in most systems. Changes in vertical stratification, and its impacts on turbulent mixing, control light exposure and nutrient fluxes through the pycnocline impacting phytoplankton biomass, structure, seasonal dynamics, and taxonomic composition (Winder and Sommer, 2012). Increases in water temperature increase both light-saturated and light-limited rates of photosynthesis (Tilzer et al. 1986; Edwards et al. 2016) and can favour

species that are better adapted to warmer waters. Increases in thermal stratification that result from warming surface waters generally decrease nutrient availability, which will favour species that are adapted to lower nutrient concentrations (Falkowski and Oliver, 2007), or can migrate to maintain their vertical position (Huisman et al. 2004). Thus, stronger vertical stratification is generally thought to favour dinoflagellates, including several species that form harmful algal blooms (HABs) (Smayda and Reynolds, 2001; Kudela et al. 2010). In addition to the toxic effects that can result in direct mortality,

HABs also have been linked to hypoxia in a number of marine environments (O'Boyle et al. 2016; Griffith and Gobler, 2020).

Most HAB forming species are dinoflagellates, and there is growing evidence that blooms of dinoflagellates are increasing in frequency, magnitude and geographic extent, particularly in coastal environments (Anderson et al., 2012). Smayda (2002) proposes that the establishment of new bloom species results from a three-step process that includes regional

translocation, colonization, and the achievement of competitive dominance. He concludes that translocation is not a sufficient condition for a non-indigenous species to bloom, and that significant habitat disturbance is necessary for a new species to achieve competitive dominance (i.e., bloom). This link between habitat disturbance and bloom expansion is one reason why climate change has been hypothesized to play a key role in the overall increase of dinoflagellate blooms in coastal waters (Wells et al., 2020). In this paper we present observations of unprecedented bottom hypoxia in southern Cape

Cod Bay (CCB) that is hypothesized to result from the emergence of a new late summer bloom species—*Karenia mikimotoi.* Prior to 2017, *K. mikimotoi* had not been found in routine sampling, but has become prevalent during the late summer months. This dinoflagellate appears to be well adapted to the environmental conditions found in southern CCB, which have changed significantly over the last several decades. We will present detailed observations of water column structure, and the distribution of DO and chlorophyll fluorescence collected during the late summer of 2020 in order to highlight the physical



conditions that favour bottom hypoxia.  Long-term changes in chlorophyll fluorescence and species composition detected by
ongoing environmental monitoring in CCB will be used to put the 2020 observations into context. Finally, we will
demonstrate that CCB is experiencing dramatic changes in physical conditions during the summer months and will speculate
on the role that regional climate change has played in this apparent regime shift.

## 2 Background

CCB is located at the southernmost extremity of the Gulf of Maine (GOM) and is influenced by both the offshore
waters and by coastal waters moving through the inshore regions of Massachusetts Bay (Signell et al. 2000).  CCB
experiences significant seasonal variations in vertical density stratification, with well mixed conditions observed for much of
the late fall and winter and strong thermal stratification during summer (Geyer et al. 1992).  The south-westerly winds that
frequently occur during the summer result in upwelling along the western coast of the bay.  Phytoplankton dynamics exhibit
significant interannual variability, but there is often a late winter—early spring bloom in response to the increase in vertical
density stratification in April (Oviatt et al. 2007).  Surface chlorophyll is typically low during the summer, when surface
waters are generally depleted of inorganic nutrients (Jiang et al. 2007).  Strong mixing and destratification in the fall often
result in a secondary bloom, driven primarily by the increased vertical flux of nutrients into the surface waters (Oviatt et al.
2007).

Long-term measurements of water properties conducted with the support of the Massachusetts Water Resources
Authority (MWRA) provide a record of seasonal variability of vertical density stratification and DO in Massachusetts Bay
and CCB extending back nearly 30 years (Libby et al. 2020).  These data indicate that while there is always a decline in near
bottom DO in the late summer and early fall, the Bays have never previously experienced hypoxia prior to the 2019 event.
The lowest DO recorded in CCB before 2019 was 3.7 mg/L, and the seasonal minimum is more typically between 5 and 6
mg/L (Xue et al. 2014).  These long-term data also document a trend of warming surface waters that is consistent with the
regional trend in the GOM (Pershing et al. 2015), the implications of which are discussed below.

CCB is home to a diverse assemblage of phytoplankton species.  There are ~300 phytoplankton taxa that have been
identified in the region, with a subset of ~30 species commonly reported (Hunt et al. 2010).  Small (< 10 μm) unidentified
microflagellates are numerically dominant throughout the year.  Of the larger identifiable species, diatoms dominate through
much of the winter, spring and autumn, but give way to dinoflagellates during the summer months when strong vertical
density stratification is present.  During the summer, dinoflagellates consist primarily of small species such as *Gymnodinium*,
but larger species such as *Ceratium* also are commonly found.  In recent years, the previously undetected dinoflagellate
species *K. mikimotoi* has been detected with increasing cell densities in samples collected throughout the GOM.  *K.
mikimotoi* is a planktonic dinoflagellate species responsible for harmful blooms in coastal waters worldwide.  It grows well
in low-light conditions and is particularly well suited to exploit light-limited environments (Li et al. 2019).  It can vertically
migrate up to 2.2 m/h (Koizumi et al. 1996), forming concentrated layers within the pycnocline that are < 1m in vertical





thickness during a bloom (Dahl and Brockmann, 1989; Bjornsen and Nielsen, 1991; Brand et al. 2012). It has low nutritional value, and many phytoplankton feeders avoid ingesting this species (Schultz and Kiorboe, 2009). *K. mikimotoi* has strong allelopathy, which is enhanced when high cell concentrations are present (Riegman et al. 1996; Uchida et al.

1999). These allelopathic effects are thought to be an important mechanism that allows *K. mikimotoi* to become the dominant bloom species, despite its relatively slow growth rate.

### 3 Methods

In late summer 2019, in response to reports of significant lobster mortality in southern CCB by the local lobster fleet, Massachusetts Division of Marine Fisheries (MDMF) staff conducted an initial survey of bottom DO. On this survey,

which was conducted on September 25 and 26, 2019, bottom DO was measured with a YSI 6920 V2-2 data sonde. Two subsequent surveys were conducted on September 30 and October 1, 2019, by the staff of the Center for Coastal Studies (CCS) and MDMF to provide greater spatial coverage. CCS staff used a YSI ProDSS Mulitparameter Meter and CastAway-CTD. In response to the hypoxic conditions reported in 2019, increased ship-based sampling was conducted during late summer and early fall in 2020. The 2020 spatial surveys collected vertical profiles with an RBR CTD equipped with a

Rinko fast-response optical DO sensor, Turner Designs chlorophyll fluorometer and Seapoint optical backscatter sensor (OBS). The CTD samples at 16 Hz and was lowered slowly by hand to the bottom providing highly resolved vertical profiles of the entire water column. Surveys in 2020 were conducted on August 31, September 3, 10, 16, 24 and October 2. At select CTD profile locations, surface and bottle samples were collected and analysed for extracted chlorophyll and dissolved inorganic nutrients. Extracted chlorophyll data were used to calibrate the vertical profiles of fluorescence and the optical DO sensor was calibrated in the laboratory.

In addition to these detailed data collected in 2019 and 2020, CCS staff have been collecting monthly water quality data at eight locations in CCB since 2006 (fig. 1). While a variety of parameters were measured beginning in 2006, collection of full water column profiles of chlorophyll fluorescence began in 2011, when surveys included a Seabird SBE-19*plus* CTD equipped with an SBE-43 dissolved oxygen sensor and WET Labs ECO-FL fluorometer. The CTD samples at 4Hz and is hand-lowered at a speed of ~ 0.5 m/s, stopping approximately 2-3m above the bottom. In addition to the CTD data, CCS collects discrete bottle samples for inorganic nutrient analysis. Bottles at both the surface and 2-3m above the bottom are collected. These samples are passed

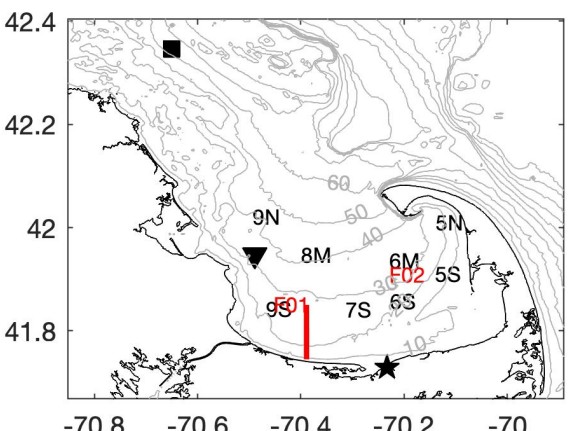

**Figure 1: Map of Cape Cod Bay (CCB) showing the locations of the CCS monthly water quality stations (black letters), the MWRA sampling locations in CCB (red letters), the location of wind data at Chapin Beach courtesy of Weather Flow, Inc. (black star), NDBC buoy 44013 (black square), MDMF Wreck of the Mars bottom temperature sensor (black triangle), and the 2020 transect shown in figure 3 (red line). Grey contours are depths in meters.**



through a 0.4 µm membrane filter and frozen for subsequent laboratory analysis on an Astoria 2 Autoanalyzer following the methods outlined in Costa et al. (2020).

The MWRA has been collecting water samples for phytoplankton identification and quantification at 34-48 locations throughout Boston Harbor, Massachusetts Bay and CCB, roughly every month since 1992. Each month, surface water samples are collected from two locations in CCB (F01 and F02). These samples are collected with a Niskin bottle, and

a 1 litre subsample is preserved with Lugol's solution. Phytoplankton samples are concentrated via gravitational setting by a factor of approximately 16:1 as described by Borkman et al. (1993), Borkman (1994), and Turner et al. (1995). A 1-mL aliquot of concentrate is transferred to a gridded Sedgwick–Rafter chamber and phytoplankton are counted using an Olympus BH-2 research microscope with phase contrast optics following methods outlined in Costa et al. (2020). Species abundance estimates are derived from these counts following the methods outlined in Hunt et al. (2010).

To place these data into the context of regional climate change in CCB, we analyse several other sources of long-term data from the region. The National Data Buoy Center (NDBC) Boston (44013) buoy has measured surface water temperature and wind (speed and direction) hourly since August 1984, providing over 36 years of nearly continuous data. Similarly, the MDMF has maintained several bottom temperature loggers in CCB beginning in July 1991. These temperature loggers sample every two hours, and here we analyse over 3 decades of bottom temperature measured at the

Wreck of Mars location in the NW corner of CCB, where the local water depth is ~ 33m (fig. 1). While this bottom sensor and the NDBC station are not co-located, they provide information about the long-term trends in both surface and bottom temperature in the region. In the analysis presented below, we take averages over the summer months (June-September), and only include years where over 90% of the data are available.

**4 Results**

The late summer 2019 surveys conducted by MDMF and CCS staff revealed a broad area of hypoxic water in southern CCB (fig.2a). The region of hypoxia extended ~20km in the along-isobath direction and at least 6km in the across-isobath direction, spanning waters depths from approximately 9 to 24m. In interviews conducted by MDMF staff, mortality in lobster traps was reported beginning on September 20, 2019, and local scallop fishers reported significant mortality in scallop trawls conducted as early as September 15, 2019. A broad survey conducted on October 8, 2019 (not shown),

following a period of relatively energetic winds from the north, found no hypoxic water in the region, suggesting that low DO bottom waters either vertically mixed or were advected offshore. Increased sampling was conducted in 2020, which demonstrated that a broad region of bottom hypoxia formed for a second consecutive summer (fig. 2b). Hypoxic conditions were first detected during an across shelf survey conducted on August 31, 2020, which found bottom DO levels below 2mg/L centred on the 20m isobath (fig. 3b). A more comprehensive spatial survey conducted on September 3, 2020,

revealed that the hypoxic water extended at least 10km in the along-isobath direction, with a general distribution largely consistent with late summer 2019. Hypoxic bottom waters persisted through September 16, 2020, but a survey conducted on September 24, 2020, after a period of sustained strong winds from the north, found bottom DO levels had increased to well above hypoxic levels.





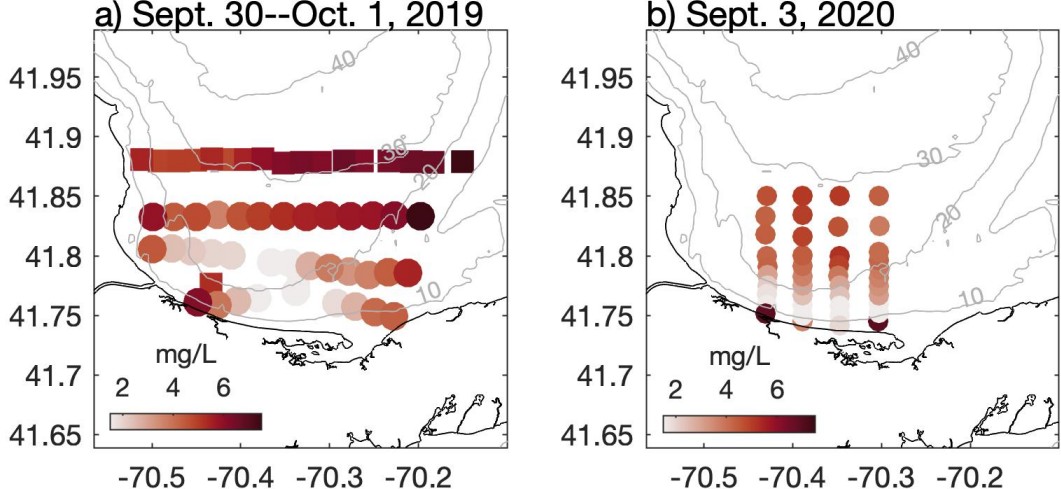

Figure 2: Bottom dissolved oxygen measured in southern CCB during late summer a) 2019 and b) 2020. In 2019 data were collected on September 30 (squares) and October 1 (circles). In both years a broad region of hypoxic bottom water developed in relatively shallow water where the thermocline intersects the seafloor. Grey contours are depths in meters.

The high-resolution sampling conducted during 2020 provided a detailed description of the distribution of DO,
chlorophyll fluorescence, optical backscatter and water column density structure (fig. 3). On August 31, 2020, strong temperature stratification was present with a horizontal density front that intersected the bathymetry (fig. 3b,c,d). The isotherms were approximately horizontal, capping a thin bottom boundary layer (BBL), that increased in height with increasing depth offshore. Low oxygen water was confined to the thin BBL, with the lowest oxygen levels roughly coincident with the thinnest BBL heights associated with the bottom density front. A sub-surface chlorophyll maximum
(>100 µg/L) was present in the pycnocline at a depth of roughly 20m. The August 31st survey followed a period of moderate winds from the north (fig. 3a). A second survey on September 3, following a period of moderate winds from the south, demonstrated that the low DO water was still present, but had been advected shoreward with the lowest bottom DO centred on the 15m isobath (fig. 3e). The upwelling associated with southerly winds pulled colder bottom waters onshore and increased the thickness of the BBL. Weak southerly winds continued for most of the next week, providing sustained
upwelling conditions, and the centre of hypoxic water had advected even closer to shore on September 10 (fig. 3h). Bottom DO levels in water depths between 20 and 30m increased and the total volume of hypoxic water decreased relative to the prior survey (table 1). The survey conducted on September 16, 2020, followed a period of moderate northerly winds, and a strong bottom density front had formed during these downwelling-favourable conditions, with a much thinner BBL than was observed 6 days prior (fig. 3k). Following this northerly wind event, the overall volume of hypoxic water increased by
roughly a factor of 2, as compared to the previous survey. Chlorophyll fluorescence was concentrated in a much thinner layer following winds from the north (e.g., Aug. 31 and Sept. 16), and a much sharper thermocline was observed. A strong



**Figure 3: a)** North-South component of wind measured at Chapin Beach. Panels on the left (b,e,h,k,n) show across-shelf transects of dissolved oxygen, panels in the centre (c,f,i,l,o) show chlorophyll fluorescence and panels on the right (d,g,j,m,p) show optical backscatter measured during a series of cruises in late summer 2020. For each transect, temperature contours (1°C interval) are shown. In the top panel the vertical red dashed lines indicate the approximate timing of each cruise. Note that Aug. 31 and Sept. 16 followed downwelling winds (positive) and Sept. 3 and 10 were during a period of prolonged upwelling-favourable winds (negative). See figure 1 for location of transect and wind measurements.





and prolonged northerly wind event in late September largely mixed the water column and no hypoxia was observed during a survey on September 24, 2020 (fig. 3n).

During all cruises, optical backscatter was highest in sub-pycnocline waters (fig. 3), and there was a statistically significant (p<0.01) positive correlation between optical backscatter and chlorophyll fluorescence (table 1).  These data suggest that phytoplankton in the water column were the primary component of the suspended particulate material detected by the OBS.  However the nature of the relationship between the OBS data and fluorescence changed between the cruises, and the slope of the linear regression between these two quantities decreased by more than a factor of 2 between the

September 10 and September 16 surveys.  This decrease in slope is consistent with the wanning stages of a bloom, where the suspended particulate matter is composed primarily of dead or weakly fluorescing cells.  The optical backscatter data suggests that these cells are found almost exclusively below the thermocline, and may be the primary source of the biological oxygen demand (BOD) that is depleting bottom waters of DO.

       The across shore surveys highlight that in CCB winds from the north drive downwelling and winds from the south

drive upwelling.  This local response is most likely driven by the upwelling/downwelling dynamics of the broader New England shelf to the north, which propagate along-shelf as a coastally-trapped wave (Allen, 1976).  The impact of upwelling/downwelling on the density structure can be seen more clearly by comparing individual profiles collected on September 10$^{\text{th}}$ (after upwelling) and September 16$^{\text{th}}$ (after downwelling) from the same mid-shelf location (fig. 4). Although winds from the south and the resulting upwelling increased the overall top-to-bottom temperature difference, the

resulting thermocline was much more diffuse, with a maximum local temperature gradient in the thermocline of ~ 2 °C/m. In contrast, after downwelling conditions, the overall top-to-bottom temperature difference was reduced, but the thermocline was both locally more intense (> 10 °C/m) and closer to the seabed.  These trends are not limited to this one station, but are reflected in the across-shelf average of all stations (table 1).  On average, upwelling conditions (Sept. 3 and 10) increase the mean temperature gradient, but downwelling winds (Aug. 31 and Sept. 16) increase the maximum value of the temperature

| Survey Date [2020] | Preceding wind forcing | Mean BBL Height [m] | Normalized Hypoxic Volume [m³/m] | Mean ∂T/∂z [°C/m] | Max. ∂T/∂z [°C/m] | Correlation OBS --Fluorometer [r] | Slope OBS --Fluorometer [µg/L/NTU] |
|---|---|---|---|---|---|---|---|
| 8/31 | Downwelling | 4.23 | 11400 | 0.28 | 3.45 | 0.80 | 25.3 |
| 9/03 | Upwelling | 8.09 | 19760 | 0.39 | 2.90 | 0.83 | 21.9 |
| 9/10 | Upwelling | 11.47 | 11970 | 0.56 | 3.00 | 0.91 | 27.3 |
| 9/16 | Downwelling | 6.09 | 27930 | 0.18 | 4.92 | 0.52 | 12.4 |
| 9/24 | Downwelling | 18.43 | N/A | 0.07 | 0.98 | 0.84 | 6.4 |

**Table 1.  Summary of conditions for 2020 cruises.  Wind forcing is based on the mean north-south component of the wind over the preceding 1.5 days (~inertial period) prior to the survey.  Mean winds from the south are considered upwelling and mean winds from the north are considered downwelling.  Only data from the central transect shown in figure 1 are reported.  Mean BBL height, mean ∂T/∂z and max ∂T/∂z are the average values for all profiles collected on the transect (n~11).  BBL height is estimated from vertical location where maximum value ∂T/∂z is observed.  The normalized hypoxic volume is calculated as the vertical and across-shelf extent of water with DO concentration < 3 mg/L and is normalized by distance in the along isobath direction (e.g. units of area).**

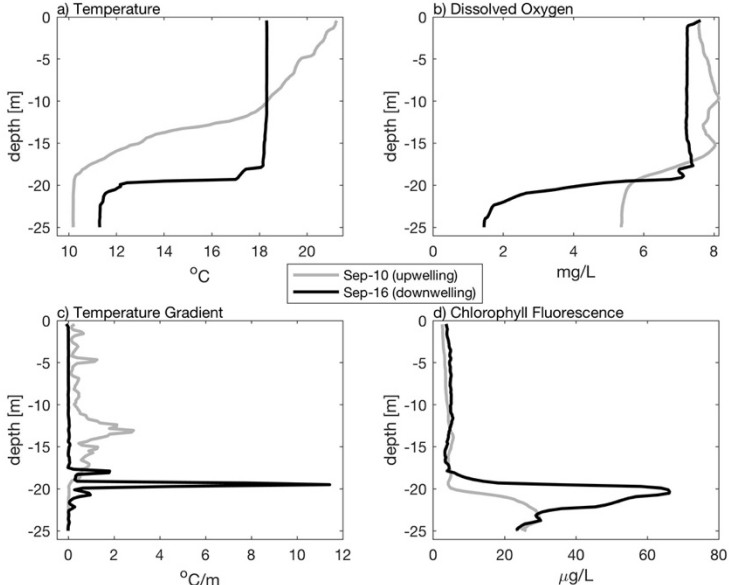

**Figure 4: Vertical profiles from the mid-shelf (~41.82$^{o}$N) collected during late summer 2020 including a) temperature, b) dissolved oxygen, c) vertical temperature gradient, and d) chlorophyll fluorescence highlighting the difference following upwelling (Sept-10 grey line) and downwelling (Sept-16 black line) conditions. Note that even though the top-to-bottom temperature difference is greater following upwelling, downwelling results in a stronger temperature gradient, thinner bottom boundary layer, lower dissolved oxygen and higher sub-surface chlorophyll fluorescence.**

gradient in the thermocline and decrease the BBL height. In these data, a discrete sub-surface chlorophyll maximum was associated with a sharper and more intense thermocline following downwelling, and low DO bottom waters were associated with a thin BBL. Thus, moderate downwelling events may result in conditions that are more favourable to both sub-surface phytoplankton production and bottom hypoxia.

A notable feature of the 2020 data is the sub-surface chlorophyll maximum that is seen in the across-shelf surveys.

To put these data into context, we compare them to the longer time series of chlorophyll fluorescence collected by CCS in CCB, which covers the period 2011-2020. The data from 2019 and 2020 are remarkable because they have some of the highest fluorescence values observed during the 10-year record, and these values are often found in discrete sub-surface maxima (fig. 5). For example, in September 2020 a sub-surface maximum with chlorophyll fluorescence ~ 100 μg/L was observed at station 9N (fig. 5g). Prior to 2019, the highest value recorded anywhere in the water column in September at this

station was ~8 μg/L. While no CCS data were collected in August 2019, the August 2020 bay-wide, eight-station average of depth-integrated chlorophyll fluorescence was the highest ever recorded. Both September 2019 and 2020 had more than 3 times the integrated chlorophyll, on average, than the September mean prior to 2019 (fig. 6a). This dramatic increase in the late summer integrated chlorophyll values for 2019-2020 is likely a key contributor to the occurrence of hypoxia during those years. If we convert the average Aug-Sept depth-integrated chlorophyll to carbon biomass, using a carbon-to-

chlorophyll ratio of ~60 (see below), the average values for the periods 2011-2018 and 2019-2020 are 225 and 745



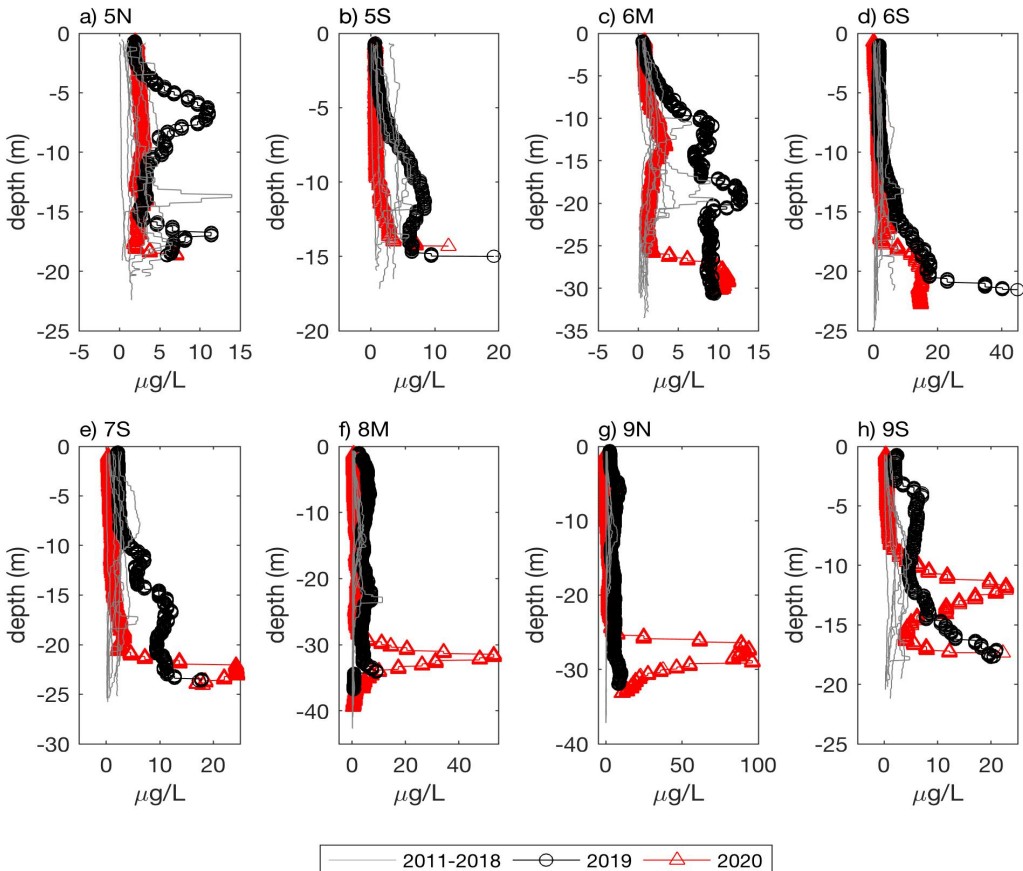

**Figure 5. Vertical profiles of chlorophyll fluorescence from the September CCS cruises for all eight CCB stations (see fig. 1 for locations). Data from 2011-2018 are plotted as grey lines and 2019 and 2020 as circles and triangles, respectively.**

mmolesC/m$^2$, respectively. If we further assume that all of this biomass is respired in a BBL that is ~5m thick, this represents a BOD of 45 and 149 mmolesO$_2$/m$^3$ (1.4 and 4.8 mg/L), respectively. Given that bottom summer DO levels typically average 156-188 mmolesO$_2$/m$^3$ (5-6 mg/L), this dramatic increase in chlorophyll biomass readily explains the switch from normoxic to hypoxic conditions given the major increase in BOD starting in 2019. It is important to note that it

doesn't matter whether the respiration of this excess organic matter occurs in the sub-pycnocline water column or the benthos. Both will deplete bottom waters of DO when a strong thermocline is present. The 2020 survey data show elevated levels of optical backscatter below the thermocline that are likely dominated by post bloom phytoplankton cells, which are the most likely source of BOD in the BBL.

The anomalously high chlorophyll fluorescence in 2019 and 2020 was accompanied by anomalously low bottom

nutrients (fig. 6b). Bottom nutrient data collected by CCS for the period 2011-2020 were used to estimate the total integrated inorganic nitrogen (NO$_2^-$ +NO$_3^-$ + NH$_4^+$) within the BBL by multiplying the bottom nutrient concentration by the

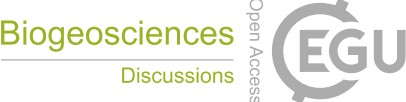



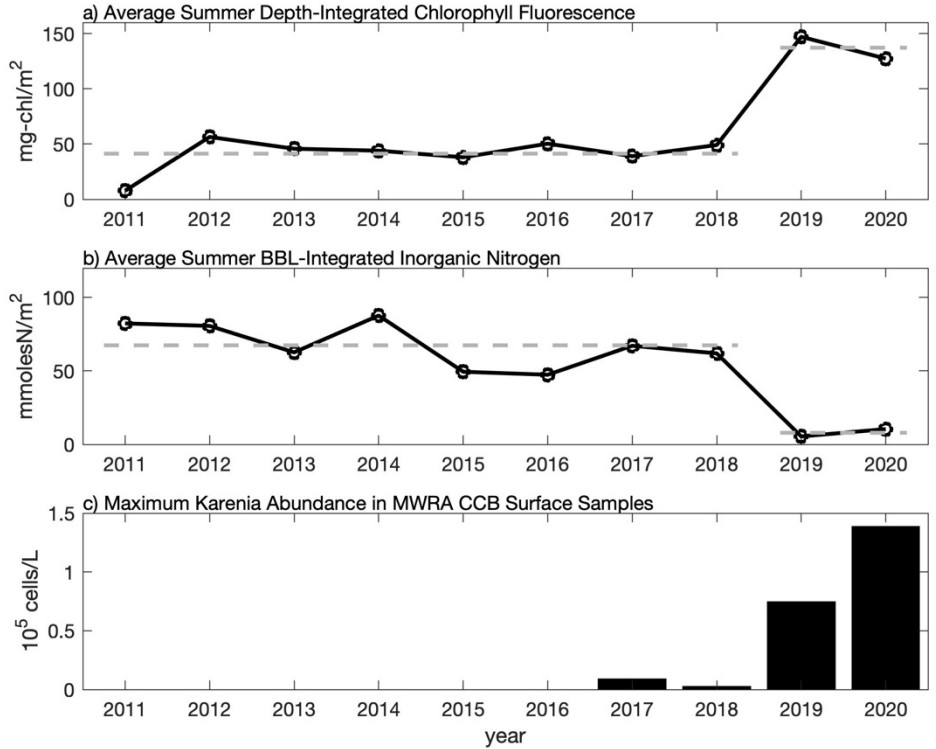

**Figure 6: Time series of the average summer a) depth-integrated chlorophyll fluorescence from CCS surveys, b) bottom boundary layer (BBL) integrated inorganic nitrogen from CCS surveys, c) maximum abundance of *K. mikimotoi* from MWRA late-summer (August and September) surface samples in CCB (F01 and F02). In a) and b) each point represents the average over the August and September CCS cruises and over all eight CCB stations. Average values for 2011-2018 and 2019-2020 are shown with the thick dashed line highlighting that the 2019-2020 data differ by more than a factor of 3 from the 2011-2018 average. In c), maximum abundance is based on the highest value reported during late summer at either F01 or F02 (see table S1 for data).**

estimated height of the BBL. BBL height was estimated from vertical CTD profiles as the first location above the seabed where the vertical temperature gradient exceeds 1°C/m. Prior to 2019, the bay-wide (eight station average) BBL-integrated nutrient concentration averaged over August and September was ~67 mmolesN/m$^2$. This value drops by more than a factor

of 8, on average, for 2019-2020 (~8 mmolesN/m$^2$) consistent with the large increase in integrated fluorescence. There is a significant ($p<0.05$) negative relationship ($r = -0.89$) between depth-integrated chlorophyll and BBL-integrated nutrients, consistent with increased nutrient consumption resulting in higher chlorophyll biomass. The significance of this relationship is clearly driven by the anomalously high chlorophyll, and anomalously low nutrients, in late summer 2019 and 2020. The slope of the linear regression between the integrated nutrients and chlorophyll is -1.32 mg-chl/mmoles-N. If nutrient uptake

is consistent with the Redfield ratio, this analysis implies a carbon-to-chlorophyll ratio of ~ 60 mg-C/mg-chl, which falls well within the range of reported values (e.g., Sathyendranath et al. 2009).



This relatively abrupt shift in sub-pycnocline nutrient utilization is consistent with a significant physical alteration of the late summer phytoplankton niche in CCB, which we hypothesize to have resulted in the recent emergence of a new phytoplankton species—*K. mikimotoi*. The MWRA has been sampling and quantifying phytoplankton species since 1992 at

~40 stations throughout the region. Prior to 2017, *K. mikimotoi* had never been found in any sample collected as part of this monitoring program. In September 2017, *K. mikimotoi* was first detected in samples collected in CCB (F01 and F02, see fig. 1 for locations). Low levels (~2500 cell/L) of *K. mikimotoi* were again detected in summer 2018, before increasing by over an order of magnitude in late summer 2019 and 2020 (fig. 6c and table S1). Maximum counts of *K. mikimotoi* at station F01 in late summer surface samples were 74,676 and 138,559 cell/L in 2019 and 2020, respectively. The MWRA only conducts

phytoplankton counts on surface samples collected from CCB. However, one bottom sample was collected from station F01 on August 31, 2020 and the *K. mikimotoi* count exceeded 771,000 cell/L, accounting for over 70% of the identifiable species at this location. This was the same day that the across-shelf survey documented an intense sub-surface fluorescence maximum (fig. 3c).

## 5 Discussion

The data presented above indicate that increased sub-surface chlorophyll production during late summer in 2019 and 2020 has provided the increased biomass to deplete bottom oxygen to hypoxic levels. The CCS nutrient data demonstrate that in the eight years prior to 2019 there were ample sub-pycnocline nutrients in CCB to support higher productivity, yet late summer sub-surface blooms of similar magnitude did not develop. We hypothesize that the intense sub-surface blooms that were observed in late summer 2019 and 2020 were *K. mikimotoi*—a dinoflagellate species that had

not been detected in the region prior to 2017. The emergence of this new species is consistent with a physical environment that has undergone significant changes. As we will discuss below, these changes include warmer waters, increased thermal stratification and important shifts in summer wind direction. These changes potentially have resulted in an environment that: 1) favours enhanced sub-surface phytoplankton production by motile species, and 2) is more physically susceptible to hypoxia.

### 5.1 Emergence of *K. mikimotoi*

With limited data on species composition, we can only speculate as to what species contribute to the sub-surface layers of high fluorescence observed in late summer 2020. The only sub-surface sample collected for phytoplankton identification during late summer 2020 had the highest *K. mikimotoi* counts ever observed in CCB. This sample was collected on the same day (Aug. 31, 2020) as the data shown in figure 3c, just west of the end of this transect. During this

survey, chlorophyll fluorescence was extremely high (>150 µg/L) and almost exclusively contained in a very thin (~2m) layer at the base of the thermocline. This intense sub-surface chlorophyll maximum is consistent with a species that is well adapted to growth in low light conditions and that can vertically maintain its position in the water column to take advantage of sub-pycnocline nutrients. *K. mikimotoi* is particularly well adapted to exploit low light environments (Honjo, 2004) and can vertically migrate at rates up to 2 m/hr (Koizumi et al., 1996). Under stratified conditions migration ceases and it is

often observed in very thin layers within the thermocline (Brand et al., 2012). The recent emergence of late summer hypoxia



also is broadly consistent with the establishment of *K. mikimotoi* as a late summer bloom species. Hypoxia has been attributed to blooms of *K. mikimotoi* in regions around the world including Japan, China, Norway, Scotland, and Ireland (Tangen, 1977; Matsuyama, 2008; O'Boyle et al., 2016; Li et al. 2019). Oxygen utilization studies indicate that most of the potential BOD from decaying *K. mikimotoi* is realized in the first five days of bloom collapse (O'Boyle et al., 2016). This

rapid depletion of DO following a bloom results from the highly labile nature of *K. mikimotoi* cells. *K. mikimotoi* has an athecate cell that lacks an armored wall and is easily ruptured, allowing quick release of organic carbon for bacterial respiration (Brand et al., 2012).

**5.2 Regional Climate Change in Cape Cod Bay**

The emergence of a new late summer bloom species is a strong indication that CCB has undergone significant

habitat alteration. Consistent with Smayda's (2002) "bloom paradigm," we hypothesize that changes to the physical environment in CCB have allowed *K. mikimotoi* to bloom in late summer. Waters in the GOM are warming more rapidly than nearly any other region in the world's oceans (Pershing et al. 2015). Surface temperature data from the NDBC Boston buoy (44013) show that surface waters are warming even more rapidly in CCB, and 2019 and 2020 were among the warmest summers recorded over the past 36 years (fig. 7a). Both light-saturated and light-limited growth rates in phytoplankton are

sensitive to temperature (Edwards et al. 2016), and the warming of CCB may have increased overall levels of primary production. However, it seems unlikely that the dramatic increase in chlorophyll fluorescence in later summer 2019 and 2020 can be explained simply by increased primary production without significant changes to the late summer phytoplankton species composition. Species transitions are expected to occur in association with increases in temperature (Eppley, 1972; Norberg, 2004) and the emergence of *K. mikimotoi* may be the result of increased summer water temperature. *K. mikimotoi*

can grow over a wide range of temperatures, but the optimal range for blooms is thought to be 20-24°C (Baohong et al. 2021). Prior to 2015, water in excess of 20°C at the Boston buoy (44013) averaged roughly 6 days a summer. However, during the last three summers (2018-2020), surface temperatures have exceeded this threshold for more than 40 days (fig. 7b).

Bottom waters at the MDMF Wreck of the Mars location (depth ~33m) also demonstrate significant increases in

temperature. However, during the summer months surface water temperatures at the Boston buoy are increasing at nearly twice the rate as bottom waters in northern Cape Cod Bay (at the Wreck of the Mars site), over the period 1991-2020. While the data collected at the Boston buoy and the Wreck of the Mars are not co-located, the long-term trends at these two sites are consistent with increasing levels of thermal stratification (fig. 7c). Increased vertical density stratification reduces vertical mixing and the associated nutrient fluxes, favouring motile species such as dinoflagellates that can maintain their vertical

position (Winder and Sommer, 2012). Thermal stratification also is significantly impacted by wind forcing. As demonstrated in figure 4 and table 1, moderate downwelling-favourable winds result in a deep and sharp thermocline, and a thin BBL. In contrast, winds from the south (upwelling) generally result in a more diffuse thermocline with a thicker BBL. At the Boston buoy (44013) the likelihood of summer winds from the northeast stronger than 7.7 m/s (15 knots) has increased by nearly 10% per decade (fig. 7d). This increase in stronger winds from the northeast is largely at the expense of



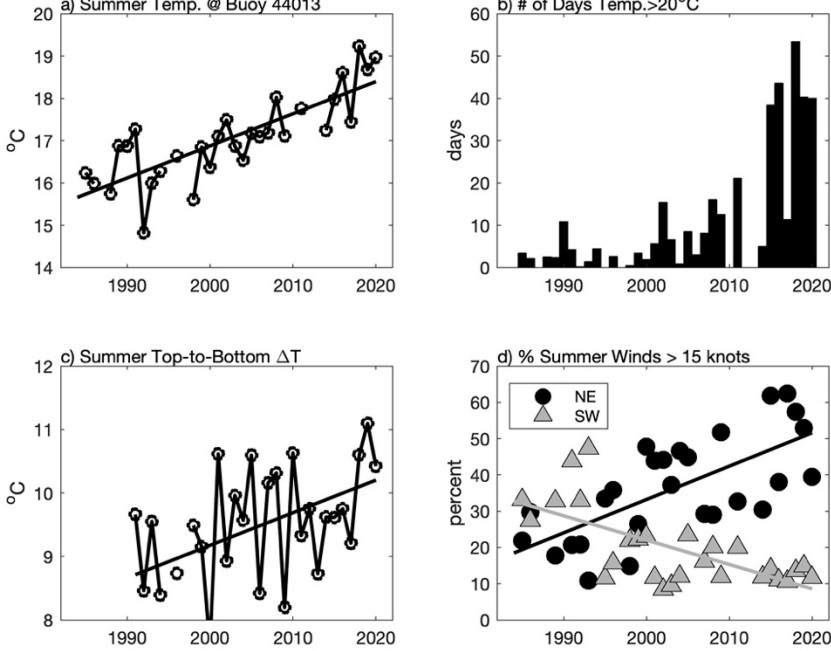

**Figure 7. Changes in summer (June—Sept) conditions in CCB including: a) mean surface water temperature at the NDBC buoy 44013, b) total number of days where surface water temperature exceeded 20°C at the NDBC buoy 44013, c) temperature difference between NDBC buoy 44013 (surface) and MDMF Wreck of the Mars (~33m depth), d) percentage of time that summer winds in excess of 15 knots (7.7 m/s) are from the NE (black circles) and from SW (grey triangle), and. Solid lines in a, c and d are linear regressions, all of which are significant at p<0.05.**

stronger winds from the southwest, which have become much less common during the summer months. This significant

increase in downwelling winds during late summer is likely to enhance deeper and stronger near bed thermal stratification—

an environment that may favour a light-adapted motile dinoflagellate species such as *K. mikimotoi*. In 2020, the highest

chlorophyll fluorescence was observed when there was a sharp well-defined thermocline; conditions that were observed

following downwelling-favourable winds (August 31 and September 16). Upwelling conditions resulted in not only a more

diffuse thermocline, but a more diffuse vertical distribution of fluorescence as well.

       Blooms of the genus *Karenia* are commonly found in coastal regions with strong horizontal density fronts, and it

has been suggested that physical convergence is important to bloom formation (Brand et al. 2012). The allelopathy of *K.*

*mikimotoi* increases significantly at high cell concentration (Li et al., 2019), so convergent physical processes may give this

relatively slow growing species a competitive advantage. Downwelling-favourable winds drive convergence in bottom

Ekman transport, which act to generate a bottom density front (Allen and Newberger, 1996). We observe a strong bottom

temperature front following downwelling conditions in CCB (e.g., fig. 3b and 3k), consistent with strong physical

convergence in this region. The significant increase in moderate downwelling conditions during late summer may have

provided a mechanism that has helped physically concentrate *K. mikimotoi,* allowing it to bloom. In contrast, upwelling-





favourable conditions generate a surface front and bring nutrients from depth to the surface—conditions that are likely to
favour faster growing, immotile species. Thus, the long-term shifts in summer wind patterns, combined with increases in
water temperature may have resulted in an environment that is much more conducive to blooms of this new species. It is
worth noting that for large parts of the GOM southwest winds are upwelling-favourable, and the long-term reduction in
summer upwelling conditions most likely has contributed to the unprecedented warming of surface waters in the region.

**5.3 Physical Susceptibility to Hypoxia**

330         The overall increase in thermal stratification combined with the significant increase in downwelling-favourable
summer winds may have resulted in an environment that also is more physically susceptible to hypoxia. The data collected
in 2020 demonstrate that hypoxia develops where strong thermal stratification intersects the seafloor, suppressing vertical
mixing and confining the BOD to a very thin BBL (Fennel et al. 2016). The thinnest BBLs and strongest local thermoclines
in 2020 were observed following moderate downwelling conditions (e.g., fig. 4), which helps explain why the overall
volume of hypoxic water was reduced following upwelling conditions and increased following downwelling (e.g., fig. 3).
Irrespective of whether the remineralization of organic matter occurs primarily in the water column or via benthic
respiration, a thin BBL effectively concentrates BOD over a smaller volume of water, resulting in a more rapid drawdown of
bottom DO. Upwelling brings bottom waters into shallower water where greater vertical mixing can take place, potentially
ventilating hypoxic bottom waters. We hypothesize that more frequent downwelling events interrupt this process and
promote an environment that is more susceptible to hypoxia due to intense near bed stratification. Even though strong
downwelling winds eventually mixed the water column ending hypoxia in both 2019 and 2020, long-term increases in
moderate downwelling-favourable winds during the summer, combined with increased overall levels of thermal
stratification, may enhance hypoxia by favouring a deep thermocline and thin BBL. The fact that such a thin BBL can be
maintained in southern CCB is likely related to the fact that it is a semi-enclosed embayment, which limits the near bed
currents and hence the bottom stress. Along open coastlines, the along-shelf bottom stress evolves to balance the along-shelf
wind stress to first order (Lentz and Fewings, 2012). However, in CCB the presence of the shoreline prevents this
acceleration, and the wind stress can be balanced by the sea-level set-up to first order, significantly reducing the bottom
stress. This weak bottom stress, and the corresponding reduction in vertical mixing, is likely a key reason why hypoxia can
develop in this environment.

**6 Conclusion**

        For two consecutive summers, bottom waters in CCB have become hypoxic. This unprecedented occurrence
appears to be the result of a change in physical conditions that favours the growth of *K. mikimotoi*, a bloom forming
dinoflagellate linked to hypoxia in a number of other coastal environments. Increased water temperatures and thermal
stratification, combined with more frequent downwelling-favourable winds, have created an environment where strong
vertical stratification is maintained very close the seabed. This results in conditions that are more physically susceptible to
hypoxia and may allow certain phytoplankton to more effectively utilize the reservoir of deep sub-pycnocline nutrients.



These dramatic changes in CCB illustrate how the complex response to climate change can significantly alter bottom oxygen dynamics and play an important role in controlling the floral composition of a coastal marine ecosystem.

**Data Availability**

All of the data presented in this manuscript are available upon request. Please direct all inquiries to the corresponding author.

**Acknowledgements**

Funding for this research was provided by the National Oceanic and Atmospheric Association Sea Grant American Lobster Initiative grant NA20OAR4170506 and by the National Science Foundation grant OCE- 2053240. We are grateful to Marc Costa, captain of the R/V Columbia, and Noa Randall for their assistance during the hydrographic surveys. MDMF

staff members A. Boeri, C. Cano, D. Perry, B. Reilly, and S. Wilcox conducted water quality sampling in 2019. We would like to thank the Massachusetts Lobstermen's Association, and commercial lobster fishers in southern Cape Cod Bay for alerting us to the 2019 hypoxic event and communicating the conditions they were observing in both 2019 and 2020.

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
