# Peer review of "Unprecedented Summer Hypoxia in Southern Cape Cod Bay: An Ecological Response to Regional Climate Change?"

_Biogeosciences, 2022_

## Author Response (AR1)

**Reviewer #1**

General comment

Ongoing environmental changes such as warming and deoxygenation in the open and coastal oceans are altering biogeochemical cycles and ecosystems. However, a comprehensive understanding of the causes and consequences of deoxygenation in coastal areas is still lacking. The manuscript under review presents new results from a study on environmental changes recently observed in the ecosystem of the southern part of the Cape Cod Bay (CCB). The manuscript is well written and the main conclusion is justified by the presented data. (The answer to the question from the title is: yes it is.) However, I have some critical comments about the statistical analysis of the time-series data presented. Therefore, I can recommend publication only after minor revisions.

Specific comments:

- Methods: Wind speed data from two sources (Chapin Beach and NDBC buoy 44013) are presented at several places in the manuscript. But, unfortunately, I could not find any description of the wind speed data at all: How have they been measured? What was the time resolution of the measurements at Chapin Beach? How representative/comparable are the measurements from the two sites? The only reference for the source of the wind speed data from Chapin Beach ('courtesy of Weather Flow, Inc.') is given in the Figure 1 caption. So, add some text describing the wind speed data from Chapin Beach and the buoy 44013, please. And: is there no time-series available for the Chapin Beach site? (it would make more sense to analyze wind speed data which have been measured closer to the sites of the oceanographic measurements.)

We only have very limited data from the Chapin Beach site (August—October 2019 and 2020). This data came from Weather Flow Inc. and is not publicly available. The data from NDBC buoy 44013 is highly consistent with the data from Chapin Beach data and is publicly available going back to 1984. Therefore, we have removed the Chapin Beach data from the manuscript and solely use the NDBC buoy 44013 data for consistency. We have included a brief description of the how the wind speed data is measured at buoy 44013 to the methods.

- Methods: page 4, line 116: Add the details for the calibration of the DO sensor in the lab.

The DO sensor is calibrated using a two-point calibration using water that has 100% saturation (obtained using aerating bubbler) and 0% saturation (obtained by removing

all DO through the addition of sodium sulfite). Language has been added describing this process.

- P8L195 and Figure 7d: I found it confusing that throughout the text 'winds from the north' and winds from the south' are mentioned and used to explain downwelling and upwelling, but in Figure 7d winds from SW and NE are shown. I do not see the point to 'change' the wind direction for Figure 7d.

When analyzing the long-term wind data from NDBC buoy 44013, the strongest changes in summer winds are for winds from the SW and NE, so these are highlighted in Figure 7d.  As discussed below, it is not clear if upwelling/downwelling in CCB is driven locally by across isobath winds (e.g., N-S) or remotely by the larger-scale response of the New England shelf to NE-SW winds.  Moreover, we do not have enough in-situ data to determine what wind direction causes the maximal upwelling/downwelling response (N-S or NE-SW).  However, we observe a downwelling response when the winds are generally from the north and an upwelling response when winds are generally from the south and this basic classification would not change if the reference angle for our winds was rotated +/- 45 degrees.

- P8L196/197: Upwelling occurs when there are winds parallel to the coast, which won't be S or N in this case (please note: the hypoxia occur along the south coast of the CCB, not at the west coast of the CCB). The CCB seems to be a 'semi-enclosed embayment' open to the north. So, I wondering whether winds from the south may just lead to a decrease in the sea level (pushing the water away from the south coast). In turn, winds from the north just lead to an increase in the sea level (pushing water towards the south coast). I am wondering whether the authors should be more careful with their wording (i.e., down-/upwelling).

There is evidence that the upwelling/downwelling response in a semi-enclosed embayment like Cape Cod Bay is driven primarily by the response along the open coastline to the north.  Allen (1976) suggests that the upwelling/downwelling signal propagates as a coastally trapped wave, and thus may be driven by the larger scale response along the broader New England coastline.  Over this region, the coastline is orientated roughly parallel to SW-NE winds, with SW winds driving upwelling and NE winds driving downwelling.  Alternatively, there may be a local response in CCB where winds from the North-South are balanced by the sea-surface slope, which drives a geostrophic along-isobath interior flow.  This along-isobath interior flow would drive bottom Ekman transport that results in upwelling for winds from South and downwelling for winds from North.  Whether the upwelling-downwelling dynamics are driven locally or remotely is an open research question and beyond the scope of this paper.  Both processes may be important.  However, our data clearly demonstrate that

the across isobath density field responds to winds from the North and South via downwelling and upwelling, respectively.  Our intent in using the terms upwelling and downwelling is to convey the locally observed response of the density field and not represent the dynamics that are driving this response.  We have added language to state this more clearly and have included more language about the dynamics that may be driving the response we observe.

-    Figure 6: K. mikimotoi appeared under 'normal' conditions in CCB already in 2017 and 2018. Is there an explanation why K. mikimotoi has appeared in CCB at all?

We have no way knowing exactly how K. mikimotoi was introduced to CCB. There are a number of potential pathways including regional circulation and transport in ballast water.  Its first appearance in the region was in summer 2017, and it was detected at locations as far north as Casco Bay in Maine and as far south as CCB. This nearly synchronous appearance over a very large region suggests transport by regional circulation, but we cannot say definitively.  We have added language about the regional appearance of this species to the manuscript.  However, the key result is that blooms of significant magnitude have started to occur in recent years.  Consistent with Smayda's model discussed on page 2 lines 54-57, we conclude that K. mikimotoi has been able to bloom in late summer in CCB because the physical environment has been significantly modified.  This is one of the key results of the paper and we suggest that the emergence of K. mikimotoi blooms is strong evidence of significant habitat alteration.  Our hypothesis is that this is either due to increased temperature or changes in wind.  It is our view that 2017 and 2018 were not normal years.  2018 was the warmest summer on record and 2017 had the strongest NE winds during summer of any year.  These very warm waters combined with the deep stratification that results from downwelling conditions are likely what allowed this new species to start blooming.

-    P13L296/297: This statement about the data presented in Figure 7b is misleading: There is only one year (i.e., 2018) with >40 days with temperatures higher than 20°C in the period 2018-2020. But there was indeed another year (i.e., 2016) with >40 days with temperatures >20°C before that period (i.e., between 1986-2017). The authors should be more careful with their wording here. Please rephrase.

2018, 2019 and 2020 had 53.4, 40.3 and 40.0 days > 20°C, respectively.  The only other year with > 40 days was 2016 (43.6).  Perhaps this is not clear from the graph, so we can add a red dashed line at 39 days and modified the language, so this is clearer.

-    Figure 7c): I am wondering whether the significance (trend) of the linear regression is only driven by the data points from 2018-2020. Please check.

Yes, if we exclude the 2019 and 2020 data, the regression is no longer significant. The slope is still negative, but there is sufficient scatter, and with relatively few data points (n=8), the slope is not significant without these two points (p>0.1). We will make sure this is clearly stated in the text.

- Figure 7d): There are two points about the wind speed data which seem to be interesting but which are not discussed at all: (i) There is an obvious regime shift around 1995. Before 1995 the percent of the SW winds (NE winds) have been increasing (decreasing). After 1995 the percent of the SW winds show no trend with time anymore, but the NE winds show an increasing trend; and (ii) Between 2014-2020 the variability of the percent of SW winds was surprisingly low (compared to the rest of the data) which obviously corresponds very nicely with the astonishing increase in days with temp. >20°C (see Figure 7b) from 2015-2020. So, I think that there were shifts in the wind regimes in 1995/6 and 2014/5. The later one may have caused the appearance of K. mikimotoi in CCB in 2017.

There may have been a regime shift in the % SW winds after 1993 (the slope of the regression for 1994-2020 is not significantly different than zero @ p<0.05). So, the significance of the overall slope appears to be driven by differences pre/post 1993. However, this is not true for NE summer winds or for surface temperature. There is no statistical support for a regime shift around 2014/2015. In fact, both NE winds and surface temperature have increased relatively steadily over the period of record. The dramatic increase in the number of days > 20°C simply reflects this long-term increase in temperature. So, there is not statistical support to suggest that a regime shift has caused the appearance of K. mikimotoi. More likely either the steadily increasing temperatures and/or increasing summer NE winds have resulted in an environment that crossed some threshold allowing this new species to bloom.

- Are there any water current data (i.e. ADCP data) available? It would be worth to see whether circulation patterns (-> ventilation, residence time of the bottom water) in the CCB have changed as well.

There is no long term ADCP data available in CCB to our knowledge. There is ADCP data from the Gulf of Maine Ocean Observing System (GOMOOS) just southeast from Gloucester, MA (GOMOOS buoy A), but this is unlikely to capture the circulation within CCB.

- Figure 7: It would make sense to calculate anomalies (i.e. the difference between the actual value and the average value from a reference period) for the data in a) and c). This will give a more robust idea about exceptionally warm or cold years (in a) or surface-bottom differences in c).

We have changed the presentation in figures 7a and 7c to show anomalies.

-    Nutrient concentrations can be significantly affected by other inputs such as rain and groundwater discharge (rain increases groundwater discharge) as well: Are there rain data (from the meteorological station at Chapin Bay?) and/or groundwater discharge data available?

There are no data quantifying groundwater inputs of nutrients to our knowledge, nor is it straightforward to estimate inputs from precipitation.  For the greater Gulf of Maine, Townsend (1998) estimated that advective inputs of nitrogen exceeded riverine and atmospheric inputs by nearly 20 times.  While his budget did not include groundwater inputs, they are likely considerably smaller than riverine inputs.  We expect similar dynamics in CCB with advective fluxes of nutrients from deeper offshore waters far exceeding atmospheric or groundwater inputs.  Our data suggest that the intense blooms that are resulting in hypoxia in CCB are formed by a motile phytoplankton species utilizing the deep reservoir of nutrients below the thermocline.  The drawdown of sub-pycnocline nutrients in 2019 and 2020 supports this basic hypothesis.  Furthermore, these data also suggest that there were ample bottom nutrients in previous years when no blooms formed.  We conclude that these intense blooms were not primarily controlled by variations in nutrient inputs, but by changes in the physical environment that allowed these deep nutrients to be utilized.

-    Would it make sense to add a brief outlook in order to speculate about the time point when CCB might becoming seasonally anoxic and the resulting consequences for fisheries and tourism?

We are reluctant to speculate about whether or not CCB is going to become seasonally anoxic given the uncertainties in what is driving hypoxia.  Our primary hypothesis is that intense blooms of K. mikimotoi provided the biomass that ultimately resulted in hypoxia in 2019 and 2020.  While physical conditions are changing in CCB and such blooms could become more common, predicting when/if a specific species will bloom is very challenging.  There is simply too much uncertainty to say with any confidence hypoxia is likely to occur moving forward.

**Reviewer #2**

I apologize for  my delay in submitting this review. I thought this paper was clearly written and convincingly documents a complex environmental change that spans physical, chemical, and biological variables. I have a few suggestions and comments.

(1) I wonder if the N-S wind plot in figure 3, currently at the top of this figure, could be placed to the left and rotated 90 degrees? This might help the wind-events be more easily aligned with the contour data. This is just a suggestion if it works.

We tried rotating the N-S wind plot but found that this orientation was confusing and was harder to read with the text of the axes rotated.

(2) were any PAR of kd measurements made? It would be a nice addition to know how much light is available at 10- and 20-meters depth in this system where the fluorescence peaks were found.

PAR data was collected in conjunction with the chlorophyll fluorescence by CCS during their monthly cruises and these data can be used to calculate values of the light attenuation coefficient (Kd). These data span the period from 2011-2020. We analyzed these data and have included a brief description of this analysis in the manuscript. In short, there are no clear long-term trends in Kd values over this period. However, on average, 2019 had the highest Kd values in late summer (Aug and Sept) and 2020 had the lowest. It is worth noting that while both years had high integrated chlorophyll, 2020 had more pronounced deep peaks and the chlorphyll distribution in2019 was more uniform. We did find a statistically significant correlation between the euphotic depth (estimated as log[.01]/Kd ) and the shallowest depth that chlorophyll fluorescence exceeds 15 µg/L in the CCS data. This is generally consistent with blooms of a light-adapted, motile species that can maintain its vertical position where there is just enough light and is adjacent to sub-pycnocline nutrients. There are only three profiles from August or September for the period 2011-2018 where the chlorophyll fluorescence exceeds 15µg/L at a depth deeper than 5m at any of the 8 CCS stations. In contrast, there are 16 profiles for the period 2019-2020. For 15 of these 16 profiles, the depth where the chlorophyll fluorescence first exceeds 15µg/L is shallower than the estimated euphotic depth. We have included language about the PAR data and our calculation of Kd. We have also added the Kd values calculated from the profiles for 2019 and 2020 to figure 5. However, what is controlling this variability in Kd is unknown and beyond the scope of this manuscript.

(3) there are a couple of instances in the 2011-2018 fluorescence records (figure 5) where values reached comparable levels to the recent data at 5N and 6M. Did you examine if these periods were associated with comparable physical conditions as 2019-2020?

As noted above, there were only 3 cases from 2011-2018 where chlorophyll fluorescence exceeded 15µg/L at any of the CCS stations. In these three cases, the chlorophyll max did not exceed 25µg/L. In all three cases the water column was

strongly stratified, the chlorophyll max was deep (~20m) and located at the base of the thermocline, so the physical conditions were generally similar to our 2020 data. However, these profiles were isolated and in an integrated sense, the chlorophyll was much lower during these years. Bottom DO was above 5 mg/L at all stations, and we see no evidence for hypoxia, consistent with the lack of a significant bloom in these years.

---

## Author Response (AR2)

Response to Editor:

The following language has been added to the section "Data Availability"

The data collected for this project have been archived (Scully et al., 2022) and are available at https://hdl.handle.net/1912/29009. The National Data Buoy Center (NDBC) buoy 44013 data are available at https://www.ndbc.noaa.gov

The appropriate reference with doi has been added to the references section.